# Investigation on Drying Shrinkage of Basalt Fiber-Reinforced Concrete with Coal Gangue Ceramsite as Coarse Aggregates

**DOI:** 10.3390/ma18194627

**Published:** 2025-10-07

**Authors:** Shi Liu, Xiaojian Rong, Shuchao Wei, Dong Li

**Affiliations:** School of Civil Engineering, Liaoning Technical University, Fuxin 123000, China; liushi@lntu.edu.cn (S.L.); r17336331974@163.com (X.R.); W2453747704@163.com (S.W.)

**Keywords:** coal gangue ceramsite, basalt fiber, drying shrinkage, internal humidity, electrical resistivity, prediction model

## Abstract

In order to investigate the basalt fiber influences on drying shrinkage of coal gangue ceramsite concrete, specimens with varying fiber dosages and matrix strength were prepared. The drying shrinkage (DS) was compared. To elucidate the characteristics of the DS, the internal humidity (IH) and electrical resistivity (ER) were also tested. The properties of the variation in the DS, IH, and ER were verified. The correlation between the values of the DS, IH, and ES was systematically analyzed, and a prediction model of DS considering the influence of fiber dosage and coal gangue ceramsite was proposed. The results showed that the incorporation of basalt fiber can significantly reduce the DS, and the value of the DS decreased with the increment of fiber dosage. The value of the DS also decreased with the enhancement of the matrix strength. An inverse relationship existed between the variation in the IH and DS, whereas the variation in the ER demonstrated a direct proportionality with the variation in the DS. The prediction model for the basalt fiber-reinforced coal gangue ceramsite concrete was obtained by modifying the AFREM model. The values predicted by the improved AFREM model demonstrated excellent consistency with the test data.

## 1. Introduction

Coal gangue is a solid waste generated during coal production [1,2]. The large-scale open-air stockpiling of coal gangue not only occupies substantial land resources but may also cause atmospheric pollution due to spontaneous combustion [3,4]. Using it as a substitute for conventional aggregates in concrete not only promotes resource utilization but also reduces the need for natural aggregate mining [5]. Coal gangue-derived ceramsite lightweight aggregates reduce the adverse effects of carbon in the matrix while saving energy via carbon combustion during calcination. Utilizing ceramsite as coarse aggregate enhances the structure of the interfacial transition zone in concrete, thereby optimizing its pore size distribution [6]. Compared to ordinary concrete, ceramsite concrete demonstrates advantages including lighter weight, superior thermal and acoustic insulation properties, and enhanced seismic resistance [7,8,9]. Compared to geopolymer concrete, ceramsite concrete offers superior workability, higher impact resistance, enhanced fracture toughness, and a broader operating temperature range [10]. However, it also presents limitations such as lower elastic modulus, higher DS, and increased cracking susceptibility, which constrain its engineering applications [11,12]. Understanding the shrinkage characteristics of coal gangue ceramsite concrete (CGCC) and developing effective shrinkage control methods provides technical foundations for practical engineering implementation.

The influence of fiber dosage on the DS of concrete has been investigated by several researchers. Zhang et al. [13] demonstrated that fibers with high elastic modulus were more effective than those with low elastic modulus in reducing the DS of the composites. Afroughsabet et al. [14] investigated the synergistic effects of hybrid steel-polyvinyl alcohol fibers on the DS of high-performance concrete. The results showed that the steel-PVA fiber combination exhibited a pronounced positive hybrid effect on reduction in the DS of the matrix. Yousefieh et al. [15] found that the fiber incorporation effectively mitigated crack formation induced by the drying-induced shrinkage in the cementitious matrix. Due to the high elastic modulus and tensile strength, the performance of the basalt fiber-reinforced concrete has been verified by researchers.

Jiang et al. [16] conducted a series of mechanical properties tests on BFRC, and the results showed that BFRC presents high flexural strength and tensile strength, but the compressive strength increases slightly at the early age and even decreases at the late age. With the increase in BF dosage, the improvement of mechanical properties of BFRC becomes more obvious; the suitable amount of BF is about 0.3% in volume fraction. Katkhuda et al. [17] investigated the compressive strength, splitting tensile strength, and flexural strength of basalt fiber-reinforced concrete; the results showed that the optimal volume fraction of the fibers was 0.3%, beyond which the mechanical properties of the matrix were weakened. Elshazli et al. [18] investigated the influence of basalt fibers on the fresh properties, mechanical properties, durability, and corrosion resistance of concrete. The results showed that a fiber volume fraction of 0.30% was the optimal ratio, demonstrating overall acceptable performance in terms of both mechanical and corrosion-related properties. Ramesh et al. [19] studied the influence of basalt fiber on crack widths and failure modes of concrete, and the results demonstrated that basalt fibers exhibited good bonding behavior with cement. However, investigations corresponding to the basalt fiber effect on the drying shrinkage and the prediction models of concrete, especially for the lightweight concrete, are still rare.

In this study, the DS of basalt fiber-reinforced CGCC was evaluated. The DS, IH, and ER of the samples were measured over 90 days. The influence of the fiber dosage and matrix strength was verified. The variation pattern of the DS, IH, and ER of the matrix was explored, and the relationship between the variation in IH, ER, and DS was evaluated. Taking into account the combined effects of coal gangue ceramsite and fiber dosage, a modified AFREM model to predict the drying shrinkage of basalt fiber-reinforced coal gangue ceramsite concrete (BFRCGCC) was proposed. The investigation may offer valuable technical references for engineering applications of the CGCC.

## 2. Experiments

### 2.1. Materials

The basic mix proportion of the CGCC is listed in Table 1, and the water to cement ratios were set as 0.40 and 0.36, respectively. The binder was Portland cement (P·O42.5); the properties of the cement are shown in Table 2. Coal gangue ceramsite with a particle size of 5–20 mm was adopted as the coarse aggregate, as shown in Figure 1, and the properties of the coal gangue ceramsite are listed in Table 3. The fine aggregates were natural river sand with a particle size of 0–5 mm. The basalt fiber was introduced as shown in Figure 2. Through existing investigations and a preliminary test by our group, the fiber dosage for the CGCC was set as 0%, 0.1%, 0.2%, and 0.3% by volume, respectively. The physical properties of the basalt fiber are listed in Table 4. Eight groups of samples were designed as shown in Table 5.

### 2.2. Test Method

#### 2.2.1. Compressive Strength Test

Compressive strength testing was performed in accordance with the Chinese standard [20], additionally referencing the relevant clauses of International Standard ISO 1920-4 [21] and the EN 12390 series [22,23,24,25], using 100 mm cube specimens. All the samples were cured for 28 days in a controlled chamber at a temperature of 20 ± 2 °C and a relative humidity of 95%. The test was performed using a servo-controlled testing machine with a loading rate maintained at 0.6 MPa/s. The compressive strength value was obtained by Equation (1) and multiplied by a reduction factor of 0.95. The compressive strength was taken as the average of three specimens in each group.(1)$$f_{cc}=F/A$$
where *f_cc_* is compressive strength of concrete (MPa); *F* is peak load of specimens (N); *A* is bearing area of specimens (mm^2^).

#### 2.2.2. Internal Humidity Test

The IH of concrete was measured with reference to Method C (Internal Relative Humidity Method) specified in EN 13578:2003 [26]. A PVC pipe was inserted into the center of a freshly cast 100 mm concrete cube with a penetration depth of 50 mm [27,28]. A steel rod was inserted into the PVC pipe, ensuring tight contact with the inner wall to prevent slurry ingress [29]. After the samples were initially consolidated, the steel rod was slowly withdrawn, and the PVC pipe opening was immediately sealed with a rubber plug. All cubic specimens were then wrapped with plastic film to prevent moisture loss. Chindaprasirt et al. [30] and Jin et al.’s [31] studies demonstrated that external moisture and Cl^−^ can be effectively prevented from penetrating the interior by applying epoxy resin to the surface of concrete specimens. After demolding, the remaining five faces of the samples were sealed with an epoxy resin layer to ensure one-dimensional moisture diffusion within the matrix [32,33].

During the initial stage of the experiment, the temperature and humidity sensors were inserted into the PVC tubes, and the pipe openings were sealed with a flexible rubber plug. All the specimens were then transferred to a curing chamber maintained at 20 ± 2 °C with 60 ± 5% relative humidity. The IH of the sample was recorded continuously for 90 days. The arrangement of the measuring device is presented in Figure 3.

#### 2.2.3. Drying Shrinkage Test

The drying shrinkage test was conducted primarily in accordance with the Chinese standard [34], with reference to the curing and testing methods specified in EN 12390 [35]. Three prismatic specimens (100 mm × 100 mm × 515 mm) were fabricated for each mixture. The specimens were demolded after 24 h, and then transferred to a standard curing chamber maintained at 20 ± 2 °C and 95% relative humidity. At a curing time of 3 d, the prismatic specimens were placed in the modified DS test device. Both the traditional test device recommended by the standard [34] and the modified DS test devices are illustrated in Figure 4a and Figure 4b, respectively. Compared with the traditional DS test device, the improved device demonstrated the following advantages: (1) Three specimens from each group can be simultaneously mounted on the same device, facilitating observation and data recording while enabling intuitive comparison of value variations across groups. (2) The enhanced design ensures greater stability, eliminating errors induced by device vibration or environmental interference. (3) Limit plates installed on the base can effectively prevent the oscillation of the samples.

All the prismatic specimens were transferred to a curing room with a temperature of 20 ± 2 °C and relative humidity of 60 ± 5% for the DS test. The micrometer gauge readings were recorded periodically throughout the 90-day test period. The DS of the samples was determined by Equation (2) [34].(2)$$\varepsilon_{\mathrm{st}}=(L_{0}-L_{t})/L_{b}$$
where *ε*_st_ is the DS of the sample corresponding to the curing time of *t* d; *L*_0_ is the initial length of the sample (mm); *L_t_* is the length of the sample at the age of *t* d (mm); *L_b_* is the measurement distance of the sample (mm).

#### 2.2.4. Electrical Resistivity Test

Cubic specimens (100 mm × 100 mm × 100 mm) were used for the ER test. Several researchers measured the ER of the concrete by embedding copper mesh in samples [36,37]. Following the method, copper mesh (100 mm × 120 mm) was introduced to the two opposing mold surfaces before pouring the sample. The ER of the matrix was measured using the LCR meter as shown in Figure 5. The ER of the samples can be calculated by Equation (3). The mean value of the three samples serves as the ER value of the group.(3)ρ=RS/L
where *ρ* is the ER (Ω·m); *R* is the resistance (Ω); *S* is the specimen’s cross-sectional area (mm^2^); *L* is the distance of the two copper meshes (mm).

## 3. Results and Discussion

### 3.1. Compressive Strength

The 28-day compressive strength of specimens with different basalt fiber dosages is shown in Table 6. To evaluate the effect of fiber dosage on compressive strength of the CGCC, the variation and 95% confidence interval of the compressive strength are also listed in the table.

From Table 6, it can be seen that for the samples with a water–cement ratio of 0.40, the compressive strength of the BFRCGCC1-0 is 37.3 MPa; compared with BFRCGCC1-0, the compressive strength of the BFRCGCC1-1, BFRCGCC1-2, and BFRCGCC1-3 decreased by 1.6%, 4.0%, and 5.4%, respectively. Similar outcomes were also reported by Niu et al. [38] and Kizilkanat et al. [39]; the variation in the values between the mixtures is relatively small. The phenomenon indicated that the basalt fiber with low fiber dosage demonstrated a negligible effect on the compressive strength of the concrete. This is possibly because the basalt fiber, belonging to high modulus fiber, at a low volume fraction, was unable to form a load-bearing skeleton within the matrix, thus failing to enhance the compressive strength [40]. Similar findings had been reported in previous investigations regarding the compressive strength of concrete with other types of fiber [41,42,43].

### 3.2. Internal Humidity

The variations in the IH of different samples with age are presented in Figure 6.

From Figure 6, we can see that with the increment of the age, the IH of the matrix showed a gradual downward trend. The matrix’s IH progressively improved with higher fiber dosage at equivalent curing durations. For example, at the age of 28 d, compared with BFRCGCC1-0, the IH of BFRCGCC1-1, BFRCGCC1-2, and BFRCGCC1-3 increased by 3.2%, 3.8% and 4.3%, respectively. The reason may be a three-dimensional network formed by fibers, which effectively suppresses microcrack development and increases the tortuosity of moisture migration paths, thereby controlling the reduction in IH [44,45,46]. Additionally, the IH of the matrix was significantly affected by the water–cement ratio. Take the specimens with 0.1% fiber dosage as an example: at the ages of 28 d and 90 d, compared with the IH of BFRCGCC1-1, the values of BFRCGCC2-1 decrease by 9.1% and 8.0%, respectively. It may be attributed to the hydration-induced water consumption within the matrix, and the comparatively lower water–cement ratio in BFRCGCC2 made a contribution to the pronounced reduction in the IH value of the matrix [47].

In order to intuitively observe the alteration of the IH of the samples with age, Equation (4) was introduced to process the IH of the specimens [48].(4)$$DR=\left[\bar{RH_{\left(t\right)}}-\bar{RH_{\left(t-1\right)}}\right]/∆t$$
where *DR* is the IH decay rate of the specimen (%/d); $\bar{RH_{\left(t\right)}},\bar{RH_{\left(t-1\right)}}$ are the IH of the specimen at time *t* and (*t* − 1), respectively (%); Δ*t* is the temporal interval between time points *t* and (*t* − 1) (d).

The IH decay rate of the samples can be obtained according to Equation (4); the results are presented in Figure 7.

From Figure 7, it can be seen that the decay rate of the IH of the BFRCGCC was large at the initial stage (0–14 d), and gradually decreased with the increment of age, and the decay rate approached 0.1% at the age of 90 d. At the same age, the IH decay rate decreased progressively with higher fiber dosage, indicating an inhibitory effect of basalt fiber on IH variation. The reasons may be that at the early stage, the cement hydration process consumed a large amount of water, the process resulted in a large decay rate of the IH [28]; as the concrete aged, its water content decreased significantly while the increasing matrix strength impeded evaporation, leading to a noticeable reduction in the IH decay rate [49].

### 3.3. Drying Shrinkage

The variation in the DS of different samples with age is demonstrated in Figure 8.

From Figure 8, it can be seen that the DS of BFRCGCC increased rapidly at the early times and gradually slowed down in the later ages. The development of DS can be divided into three stages: Stage I (rapid growth phase), Stage II (relatively slow growth phase), and Stage III (slow growth phase). Take BFRCGCC1-1 as an example: the DS of the matrix was 523.75 × 10^−6^ at the age of 28 d and the average growth rate was 20.95 × 10^−6^/d (Stage I); during the age of 28–56 d, the DS of the specimens was 104.73 × 10^−6^ and the average growth rate decreased to 3.74 × 10^−6^/d (Stage II); during the age of 56–90 d, the value of the DS was only 41.04 × 10^−6^ and the average growth rate was only 1.21 × 10^−6^/d (Stage III).

The incorporation of basalt fibers can significantly affect the DS of the CGCC. Take the values of 28 d, 56 d, and 90 d as an example: to verify the impact of fiber reinforcement on DS of CGCC, the variation in DS was given (the values of CGCC without fiber reinforcement set as 1.0), as presented in Figure 9.

From Figure 9, we can see that the values of the DS of the matrix decreased with the increment of fiber dosage. Fiber incorporation leads to improvement in the tensile strength of the concrete matrix, which helps to physically limit shrinkage [48]. The shrinkage-induced interfacial stress at the fiber–matrix interface was transferred through the interconnected fiber network, enabling effective stress distribution and resulting in reduced matrix shrinkage deformation [50,51]. Additionally, the fibers formed an irregular spatial structure inside the matrix [52]; the fibers condensed and hardened with the surrounding grout, forming a barrier in the pores. In this way, the maximum pore size may be effectively reduced [44,53], and the number of harmful voids was further reduced, thereby the DS of CGCC was reduced significantly [54].

### 3.4. Electrical Resistivity

The ER of CGCC under varying basalt fiber incorporation levels is demonstrated in Figure 10.

From Figure 10, at the initial stage, the ER increased rapidly, and then the growth rate gradually slowed down. Take the BFRCGCC1-0 as an example: 0~28 d, the average growth rate was 10.13 Ω·m/d (Stage I); 28~56 d, the average growth rate was 4.46 Ω·m/d (Stage II); 56~90 d, the average growth rate was only 3.16 Ω·m/d (Stage III). The reason may be abundant free water in the fresh CGCC mixture; the grout with rich liquid-phase conductive ions had a large degree of freedom, resulting in a very low initial ER [55,56]. The hydration of the cement, coupled with moisture evaporation, drove rapid internal water depletion within the specimens, and then the strength of the matrix progressively enhanced as hydration products accumulated, and the flow ability of the liquid-phase conductive ions was decreased, and the ER was enhanced gradually [57,58,59].

As evidenced by the data, the ER of CGCC progressively diminished, corresponding to the rise in water–cement ratio. Compared with BFRCGCC1-0, the ER of BFRCGCC2-0 increased by 12.64%. An increment of water–cement ratio caused more continuous capillaries to appear in the cement paste, thereby reducing the ER [60]. The ER of CGCC exhibited a progressive enhancement with higher fiber dosage. This phenomenon may be attributed to that with the addition of the fibers, on the one hand, the non-conductive phase in the matrix increased; on the other hand, a higher fiber dosage reduces the number of harmful pores and increases the compactness of the concrete matrix, thereby enhancing the ER of the matrix [61].

## 4. Prediction Models

### 4.1. Prediction Model for Internal Humidity

A model of the relationship between IH and age of concrete without fiber is presented in Equation (5) [29].(5)$$H\left(t,0\right)=100-at^{b}$$
where *H*(*t*,0) is the IH of the matrix (%), 0 means the matrix without any fiber reinforcement; *t* is the test age (d); *a*, *b* are the parameters obtained by regression analysis.

The variations in the IH of BFRCGCC1-0 and BFRCGCC2-0 with age may be verified by this model, and the regression analysis results have been tabulated in Table 7.

From Table 7, we can see that the values of *R*^2^ of the specimens are larger than 0.950, which means that the IH of coal gangue ceramsite concrete can be determined using Equation (5). The variation in the IH with the increase in fiber dosage is illustrated in Figure 11.

From Figure 11, it can be seen that the relationship between the IH of the matrix and the fiber dosage was nonlinear. The influence of the fiber dosage on the variation in the IH of the concrete can be considered by Equation (6) [28]:(6)$$H\left(t,\omega \right)=K_{t}\sqrt{\omega }+H\left(t,0\right)$$
where *H*(*t*, *ω*) is the IH of CGCC with different basalt fiber dosage (%); *K_t_* is the time impact coefficient; *ω* is the fiber dosage (kg/m^3^).

The IH of the samples corresponding to the 3 d, 7 d, 14 d, 28 d, 56 d, and 90 d were adopted, and the values of *K_t_* can be obtained through the nonlinear fitting, and the results are listed in Table 8.

From Table 8, we can see that the values of *R*^2^ of eight groups were larger than 0.900, and only four groups of the *R*^2^ values were between 0.900 and 0.850. This validated that the fiber influence on the IH of the matrix followed the relationship indicated by Equation (6).

The correlation between the time influence coefficient *K_t_* and age for CGCC specimens with varying water–cement ratios is shown in Figure 12.

From Figure 12, it can be seen that the value of *K_t_* increased with the increment of age. The phenomenon proved that basalt fiber addition can effectively slow down the reduction in the IH values of the matrix. The time influence coefficient *K_t_* reflected the effect of the IH attenuation trend with age. The correlation between *K_t_* and curing duration can be described by Equation (7):(7)$$K_{t}=\alpha \times exp(-t/\beta )+\gamma$$
where *α*, *β*, and *γ* are the parameters obtained by regression analysis.

Substituting Equation (7) into Equation (6), the equation regarding the fiber influence can be obtained as presented in Equation (8):(8)$$H\left(t,\omega \right)=\sqrt{\omega }\left[\alpha \times exp(-t/\beta )+\gamma \right]+H\left(t,0\right)$$

The prediction values and the test values of the IH of BFRCGCC are presented in Figure 13, and the prediction results of RMSE and MAPE are displayed in Table 9.

The data from Figure 13 and Table 9 illustrated that there was a strong correlation between the predicted values and experimental values. Further analysis of correlations was performed through linear regression. The comparison between the test results and the predicted values is presented in Figure 14.

From Figure 14, we can see that the correlations between the test values and predicted values were larger than 0.990. The revised prediction model can be effectively employed to verify the variation in the IH of the BFRCGCC.

### 4.2. Prediction Model for Drying Shrinkage

Several predictive models have been established to estimate the DS of conventional aggregate concrete, such as the B3 model, CEB-FIP model (Model Code 90, Model Code 2010), AFREM model, ACI209.2R model, and GL2000 model [62,63,64,65,66]. Take the DS of the BFRCGCC1-0 and BFRCGCC2-0 as examples. The comparison between the predicted values by the models and the test values is shown in Figure 15. Existing studies had shown that the size of the test block reflected the moisture transport rate within concrete, serving as a critical element influencing shrinkage development [67]. Unlike conventional concrete, CGCC exhibits moisture migration both within the matrix and from the aggregates to the matrix. This dual moisture transfer makes the specimen size effect particularly critical in developing its DS prediction models. As mentioned in reference [68], the AFREM model considered the size of the components and the influence of age development on DS. Among the models, the AFREM model was frequently introduced to verify the DS of the concrete with recycled aggregates and lightweight aggregates [69,70,71].

The expression of the AFREM model [68]:(9)$$\varepsilon_{\mathrm{ds}}\left(t,t_{s}\right)=\beta_{\mathrm{ds}}\left(t,t_{s}\right)\cdot \varepsilon_{\mathrm{ds}0}$$(10)$$\beta_{\mathrm{ds}}\left(t,t_{s}\right)=(t-t_{s})/\left[\beta_{ds}h^{2}+(t-t_{s})\right]$$(11)$$\beta_{\mathrm{ds}}=\left\{\begin{array}{c} 0.007\\ 0.021 \end{array}\right.\;\begin{array}{c} \left(\mathrm{with}\;\mathrm{silicon}\;\mathrm{fume}\;\mathrm{additon}\right)\\ \left(\mathrm{without}\;\mathrm{silicon}\;\mathrm{fume}\;\mathrm{additon}\right) \end{array}$$(12)$$\varepsilon_{\mathrm{ds}0}=K\left(f_{\mathrm{cm}}\right)\times A\left(f_{\mathrm{cm}},RH_{\mathrm{ambient}}\right)\times B\times 10^{-6}$$(13)$$K\left(f_{\mathrm{cm}}\right)=\left\{\begin{array}{c} 18\\ 30-0.21f_{\mathrm{cm}} \end{array}\right.\;\begin{array}{c} \left(f_{\mathrm{cm}}\leq 57\;\mathrm{MPa}\right)\\ \left(f_{\mathrm{cm}}>57\;\mathrm{MPa}\right) \end{array}$$(14)$$A\left(f_{\mathrm{cm}},RH_{\mathrm{ambient}}\right)=72exp(-0.046f_{cm})+75-RH_{\mathrm{ambient}}$$
where *ε*_ds_(*t*, *t_s_*) is the DS of the matrix; *β*_ds_ (*t*, *t_s_*) is the development coefficient of DS over time; *ε*_ds0_ is the final DS; *t* is the age at the moment considered; *t_s_* is the age at the beginning of DS; *h* is the cross-section size of the sample, *h* = *A*_c_/*u*, *A*_c_ means the cross-sectional area of the samples, *u* means the perimeter of the cross-section; *K*(*f*_cm_) is a coefficient depending on the strength and humidity diffusion rate of the matrix; *A*(*f*_cm_, *RH*_ambient_) is the autogenously shrinkage due to the hydration; *f*_cm_ is the strength of the concrete; *RH*_ambient_ is the relative humidity of the environment; *B* is the coefficient regarding to the effect of fly ash and slag.

From Figure 15, we can see that all the predicted values were smaller than the experimental values. The existing predictive models for the traditional aggregates concrete cannot be directly applied to the CGCC. Through the predicted results, the variation in the values predicted by the AFREM model was similar to the variation in the test values. The relationship between the test values and predicted DS values of 7, 14, 28, 40, 50, 60, 70, 80, and 90 d is presented in Figure 16.

From Figure 16, it can be seen that there was a linear correlation between experimental and predicted values of DS; therefore, the AFREM model was adopted, and a magnification correction factor *M*_CG_ was introduced to modify the model. The improved AFREM model was presented in Equation (15). The comparison of the test values and the predicted values by the revised model was also presented in Figure 17.(15)$$\varepsilon_{\mathrm{ds}0}=M_{\mathrm{CG}}\times K\left(f_{\mathrm{cm}}\right)\times A\left(f_{\mathrm{cm}},RH_{\mathrm{ambient}}\right)\times 10^{-6}$$
where *M*_CG_ is the coefficient of coal gangue ceramsite.

From Figure 17, we can see that the *R*^2^ values of the two types of specimens were 0.974; this means that the revised AFREM model can be used to predict the DS of the CGCC.

The DS of the CGCC was reduced with the basalt fiber addition. Therefore, an impact factor *K*_BF_ was introduced to Equation (15) [72]. The relationship between the impact factor *K*_BF_ and the basalt fiber dosage is presented in Figure 18.

From Figure 18, we can see that *K*_BF_ decreased with the increment of age and fiber dosage, and a linear relationship was adopted:(16)$$K_{\mathrm{BF}}=\lambda_{t}\omega +P_{t}$$
where *λ_t_* is the coefficient of basalt fiber dosage; *ω* is the fiber dosage; *P_t_* is the coefficient of age.

Based on Figure 18, the values of the *λ_t_* and *P_t_* can be obtained regarding the ages of 3 d, 7 d, 14 d, 28 d, 56 d, and 90 d. The results were listed in Table 10. The variation in the *λ_t_* and *P_t_* with age can be gained by fitting, and the results are presented in Figure 19.

Figure 20 presents the comparison between experimental measurements and prediction values by the revised AFREM model of DS of the BFRCGCC.

To systematically analyze the prediction performance, indicators such as RMSE, NRMSE, and MAPE were also calculated; the NRMSE was the normalized RMSE and calculated according to Equation (17). The results were displayed in Table 11.(17)$$\mathrm{NRMSE}=\mathrm{RMSE}/(y_{max}-y_{min})$$

The results indicated that there is a good agreement between the predicted values by the revised AFREM model and experimental measurements. The revised AFREM model demonstrated that the DS of BFRCGCC decreased with the increment of basalt fiber dosage.

Whether the revised AFREM model can accurately predict the DS in other tests remains to be verified. Li et al. [50] investigated the effect of basalt fiber dosage (0%, 0.05%, 0.06%, 0.07%) on DS of BFRC. The four specimen groups were designated as plain concrete, BFRC−0.05%, BFRC−0.06%, and BFRC−0.07%, respectively. Figure 21 presents the comparison between predictions from the revised AFREM model and experimental results from Li’s investigation; the findings confirm the feasibility of the revised AFREM model for DS prediction of concrete.

### 4.3. Prediction Model for Electrical Resistivity

In order to quantitatively analyze the relationship between the basalt fiber dosage and the ER increment of CGCC, the relationship between the ER of CGCC without fiber reinforcement and the age was verified first. Figure 22 presents the fitting outcomes. Equation (18) describes the evolution of ER with age.(18)$$\rho \left(t,0\right)=mt^{n}$$
where *m* and *n* are the coefficients.

The variation in the ER of the matrix with age could be described by the exponential function. Figure 23 presents the variation in the ER with the fiber dosage.

Based on Figure 23, it can be seen that the ER increased with the basalt fiber dosage. The increment in the dosage showed a nonlinear law, which can be expressed by Equation (19).(19)$$\rho \left(t,\omega \right)=R_{\mathrm{ta}}\omega^{R_{\mathrm{tb}}}+\rho \left(t,0\right)$$
where *ρ*(*t*, *ω*) means the ER of the basalt fiber-reinforced matrix; *ω* means the fiber dosage; *R*_ta_ and *R*_tb_ are the coefficients; *ρ*(*t*, 0) means the ER of the matrix without fiber reinforcement.

The test values corresponding to 3 d, 7 d, 14 d, 28 d, 56 d, and 90 d were selected, and the values of *R*_ta_ and *R*_tb_ can be obtained by the regression analysis, and the results are listed in Table 12. Based on Table 12, the variation in the coefficients *R*_ta_ and *R*_tb_ over time can be obtained, and the results are presented in Figure 24.

From Figure 24, it can be seen that the coefficient *R*_ta_ increased with age before 56 d, and then started to decrease. The coefficient *R*_tb_ decreased with age before 14 d and then became stable. This phenomenon also reflected that the basalt fiber incorporation may contribute to the enhancement in the ER of CGCC to a certain extent, while this effect gradually weakened in the later stage.

The prediction model of ER for BFRCGCC was as follows:(20)$$\rho \left(t,\omega \right)=\left(at^{2}+bt+c\right)\omega^{k+\lambda \times exp(-t/v)}+mt^{n}$$
where *a*, *b*, *c*, *k*, *λ*, *v*, *m*, and *n* are the coefficients.

A comparative analysis of the predicted values and the test values of the ER of the matrix was presented in Figure 25.

Based on Figure 25 and Table 13, we can see that all the correlation coefficients are above 0.980, and the values of NRMSE are less than 0.05, and the values of MAPE are less than 3.00%. These mean that the predicted values of the ER of the specimens were consistent with the measured values, and the predicted models were feasible.

### 4.4. Relationship Between the IH and DS

To elucidate the correlation between IH and DS in BFRCGCC, Figure 26 presents a comparative analysis of the test values.

From Figure 26, we can see that the IH of the specimens decreased with age, and the DS of the matrix gradually increased with age, and the two indices were negatively correlated. Studies have shown that there is a nice correlation between the IH reduction and the DS of the concrete [73]. The relationship between the IH reduction and DS of CGCC with different basalt fiber dosages is presented in Figure 27.

The relationship between the IH reduction and DS can be expressed using the following Equation.(21)$$\varepsilon_{\mathrm{DS}}=A_{1}+B_{1}\ln\left(\Delta H+C_{1}\right)$$
where *A*_1_, *B*_1_, and *C*_1_ are the coefficients; Δ*H* is the reduction in the IH.

The regression results were listed in Table 14.

Based on Table 14, we can see that all the values of the *R*^2^ were larger than 0.970; the results indicated a significant linear correlation between IH reduction and DS of the samples.

### 4.5. Relationship Between the ER and DS

To elucidate the correlation between ER and DS, a comparative analysis of matrix DS and ER variations was conducted, and the results are presented in Figure 28.

Based on Figure 28, it can be seen that the ER and DS of the BFRCGCC gradually increased with age, and the two indices were positively correlated, as presented in Figure 29.

With ER as the abscissa and DS as the ordinate, the experimental results were fitted, and the relationship can be expressed by the following Equation.(22)$$\varepsilon_{\mathrm{DS}}=A_{2}+B_{2}\ln\left(\rho +C_{2}\right)$$
where *A*_2_, *B*_2_, and *C*_2_ are the coefficients.

Based on Table 15, we can see that the values of *R*^2^ of all the specimens were above 0.950; the DS of the matrix exhibited a significant logarithmic change relationship with the change in ER of the matrix.

### 4.6. Comparison of Measured Values with Those of Portland Cement Concrete

The results of this study indicate that CGCC exhibits significantly higher drying shrinkage than ordinary Portland cement concrete (OPC). As shown in Figure 30, the 28-day drying shrinkage values of BFRCGCC1-0 and BFRCGCC2-0 both exceeded 550 × 10^−6^, while the OPC shrinkage values reported in Reference [74] under similar mix proportions and curing conditions ranged between 380~500 × 10^−6^.

The rigid restraint effect of aggregate is the primary factor responsible for the difference in shrinkage. In ordinary concrete, the dense, high-elastic-modulus crushed-stone aggregate provides strong physical restraint to the matrix, effectively inhibiting its shrinkage deformation. However, the coal gangue ceramsite used in this study has a relatively low elastic modulus, resulting in a much weaker restraining effect on the shrinkage of the surrounding matrix compared to conventional aggregate, which ultimately leads to greater macroscopic shrinkage deformation.

## 5. Conclusions

Through experimental and analytical investigation, the conclusions can be summarized as follows:The addition of basalt fiber affects the internal humidity and electrical resistivity of CGCC; both the internal humidity and the electrical resistivity increased with the increment of fiber dosage.CGCC exhibits rapid drying shrinkage development during the initial stage, followed by progressive stabilization in the subsequent stage. It was found that a basalt fiber dosage of 0.3% was optimal, reducing the 90 d drying shrinkage by 20.67%.Variations in the internal humidity and electrical resistivity of the concrete serve as indicators for the drying shrinkage of the matrix. The internal humidity and the electrical resistivity may be introduced to verify the drying shrinkage of BFRCGCC.The improved AFREM model’s predicted values of drying shrinkage are consistent with the test values, providing an effective tool for predicting drying shrinkage of the BFRCGCC. The findings can serve as a theoretical basis for shrinkage control and structural design of CGCC, offering significant technical guidance for engineering applications.

Future research will focus on the following: (1) Extending the testing period to evaluate the long-term drying shrinkage behavior of BFRCGCC and further investigate durability indicators such as crack resistance, freeze–thaw resistance, and carbonation resistance, thereby providing more reliable data to support its large-scale engineering application. (2) Exploring the synergistic effects of other fiber types (e.g., polypropylene fiber, steel fiber) or the combination of basalt fiber with other materials (e.g., admixtures, nanomaterials) on the performance of CGCC, aiming to identify superior modification strategies.

## Figures and Tables

**Figure 1 materials-18-04627-f001:**
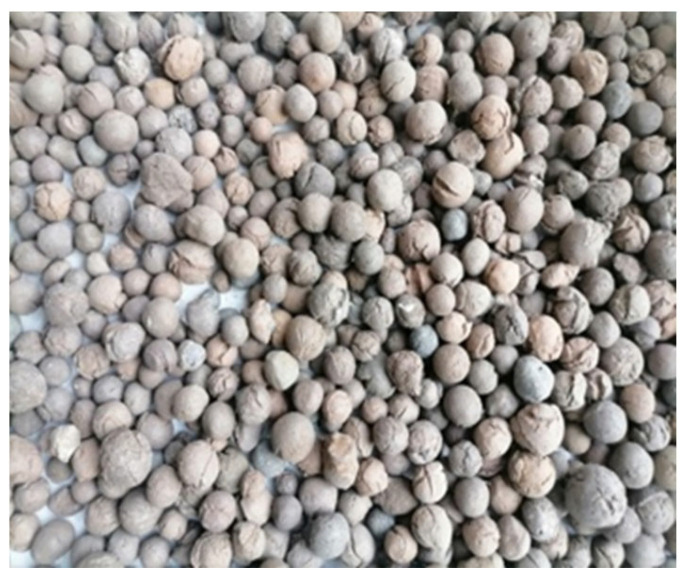
Coal gangue ceramsite.

**Figure 2 materials-18-04627-f002:**
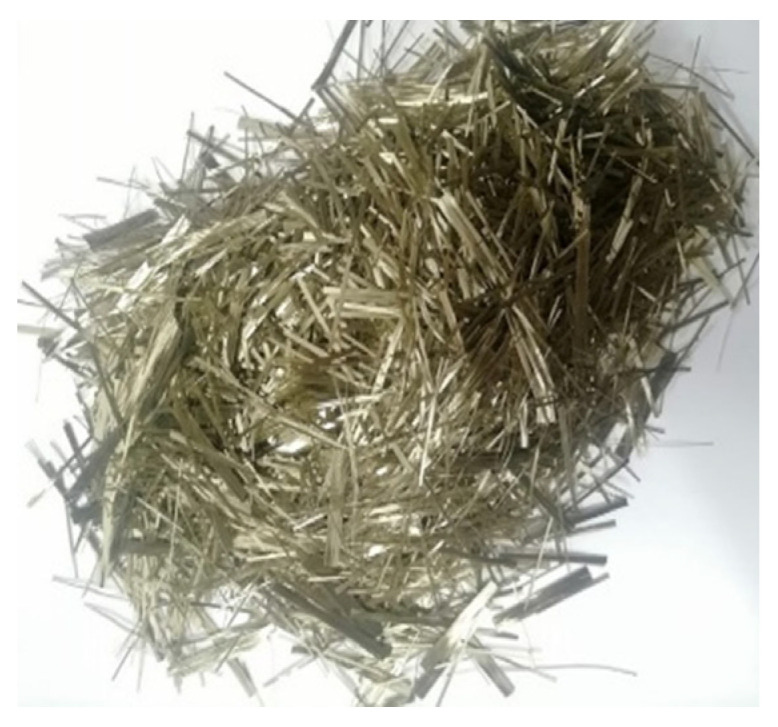
Basalt fiber.

**Figure 3 materials-18-04627-f003:**
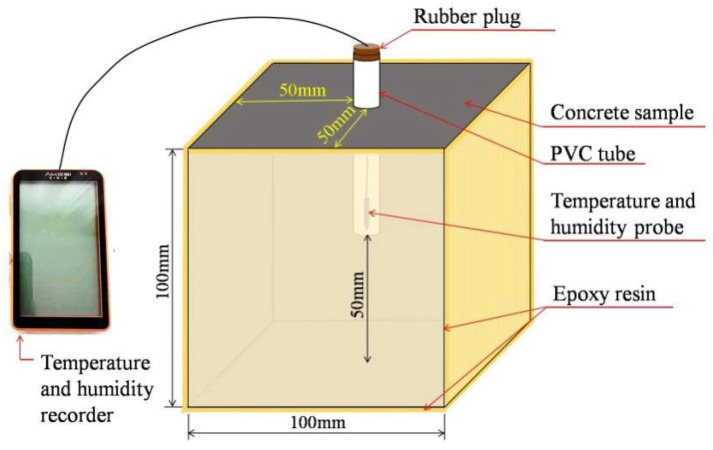
The arrangement of the measuring device for the IH test.

**Figure 4 materials-18-04627-f004:**
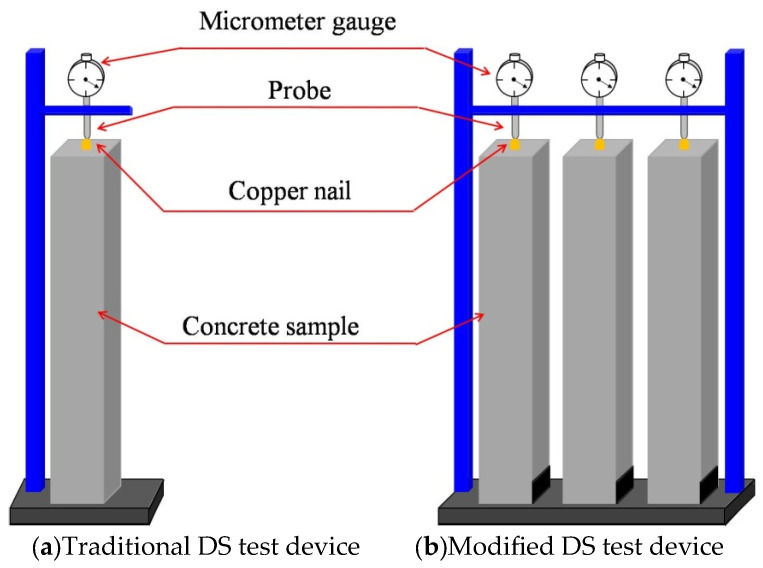
Comparison of the DS test device.

**Figure 5 materials-18-04627-f005:**
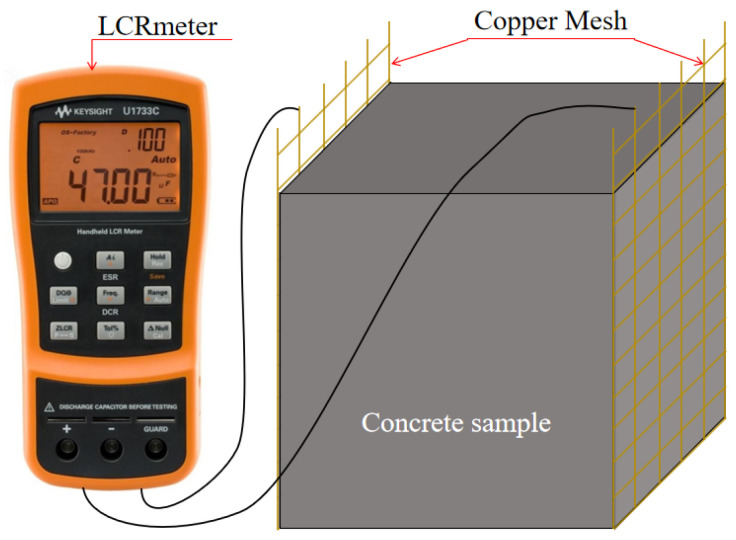
Specimens for ER testing.

**Figure 6 materials-18-04627-f006:**
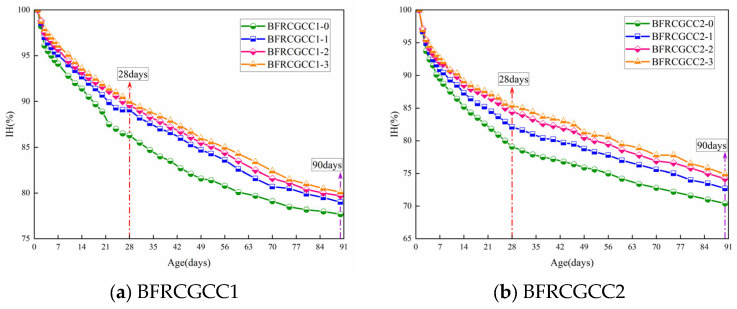
Variation of the IH of BFRCGCC.

**Figure 7 materials-18-04627-f007:**
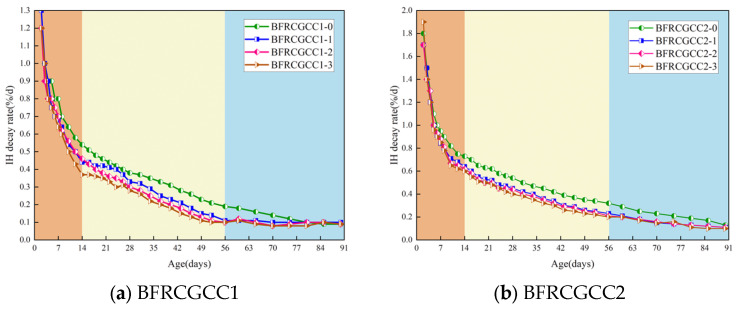
Decay rate of the IH of BFRCGCC.

**Figure 8 materials-18-04627-f008:**
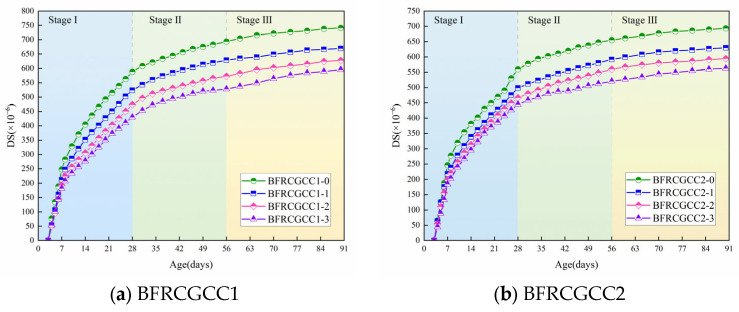
DS of the BFRCGCC.

**Figure 9 materials-18-04627-f009:**
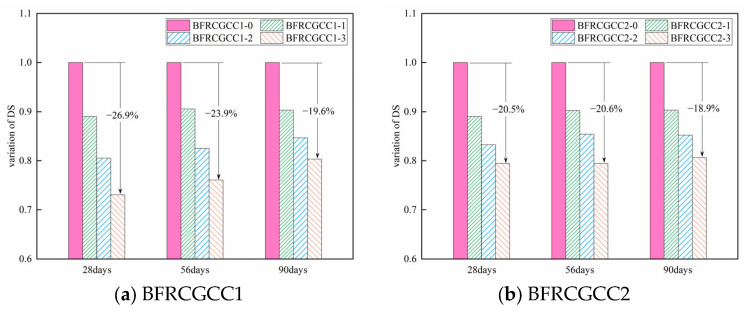
Variation in the DS of BFRCGCC.

**Figure 10 materials-18-04627-f010:**
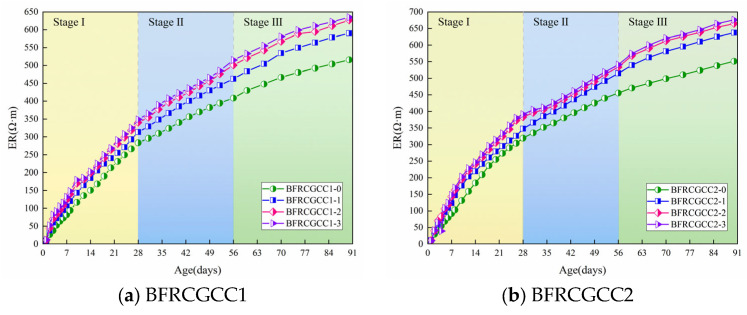
Variation in the ER of BFRCGCC.

**Figure 11 materials-18-04627-f011:**
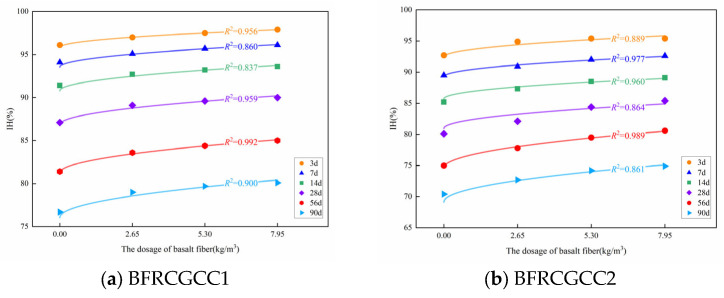
Relationship between IH and fiber dosage at different curing ages.

**Figure 12 materials-18-04627-f012:**
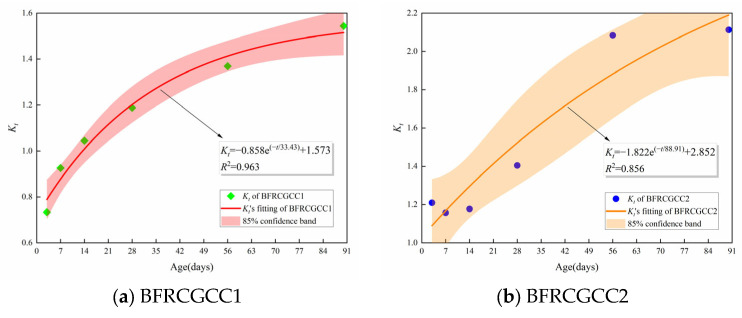
Relationship between *K_t_* and curing age.

**Figure 13 materials-18-04627-f013:**
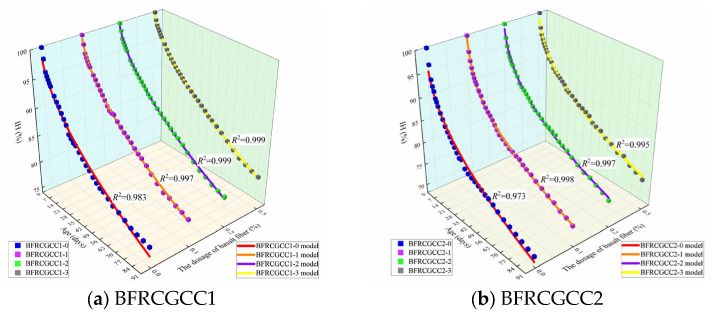
Comparison between experimental values and predicted values of the IH of the samples.

**Figure 14 materials-18-04627-f014:**
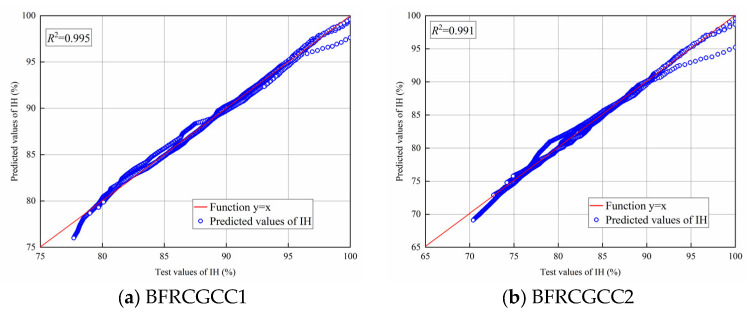
Comparison between experimental and calculated values.

**Figure 15 materials-18-04627-f015:**
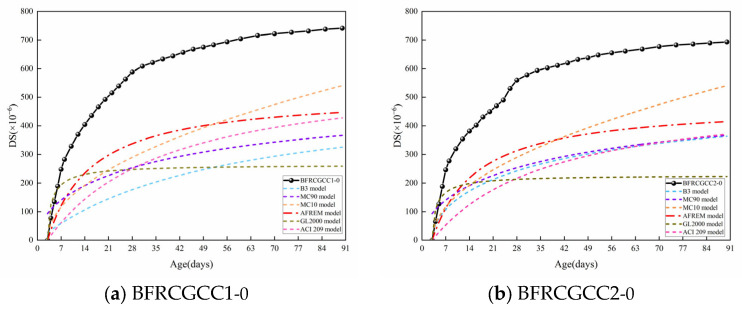
Comparison between the predicted values and test values of DS of the samples.

**Figure 16 materials-18-04627-f016:**
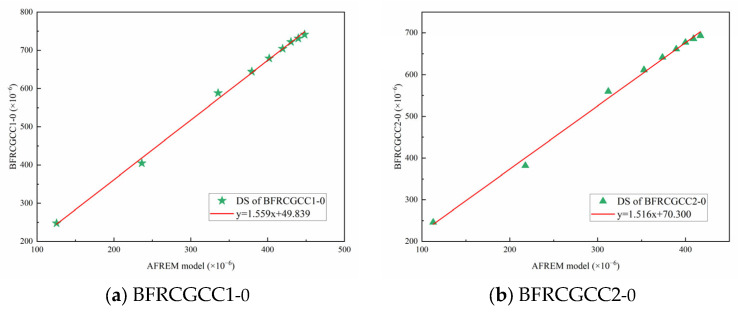
Relationship between test and predicted values of DS.

**Figure 17 materials-18-04627-f017:**
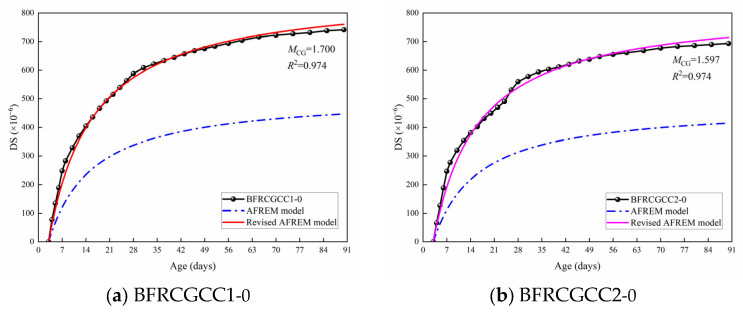
Comparison between the test values and the predicted values by the revised AFREM model.

**Figure 18 materials-18-04627-f018:**
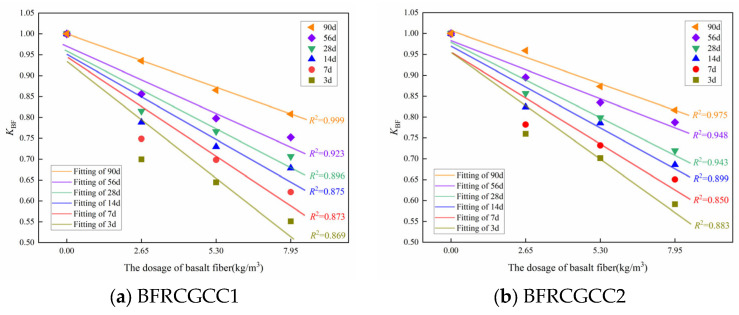
Relationship between *K*_BF_ and fiber dosage at different curing ages.

**Figure 19 materials-18-04627-f019:**
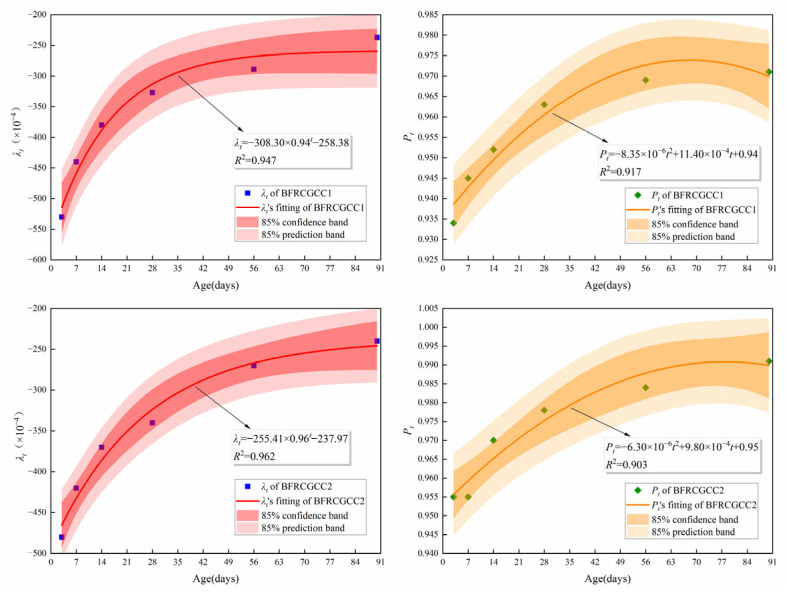
Fitting results of *λ_t_* and *P_t_*.

**Figure 20 materials-18-04627-f020:**
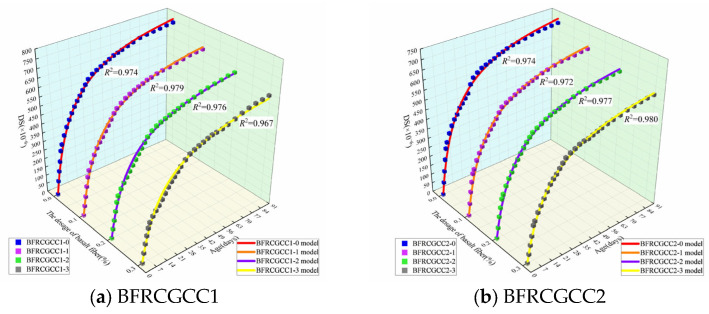
Comparison between experimental values and predicted values of DS.

**Figure 21 materials-18-04627-f021:**
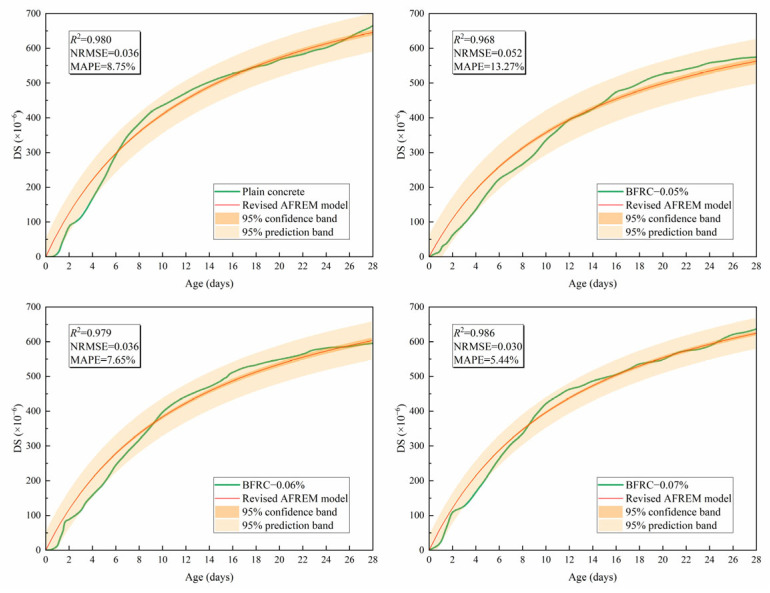
Comparison between the revised AFREM model and Li’s experimental values.

**Figure 22 materials-18-04627-f022:**
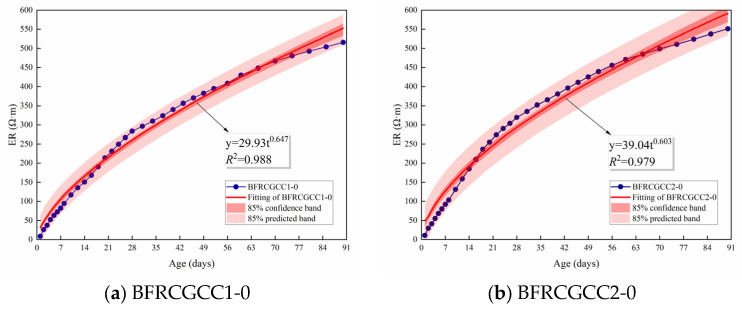
Fitting results of ER of CGCC.

**Figure 23 materials-18-04627-f023:**
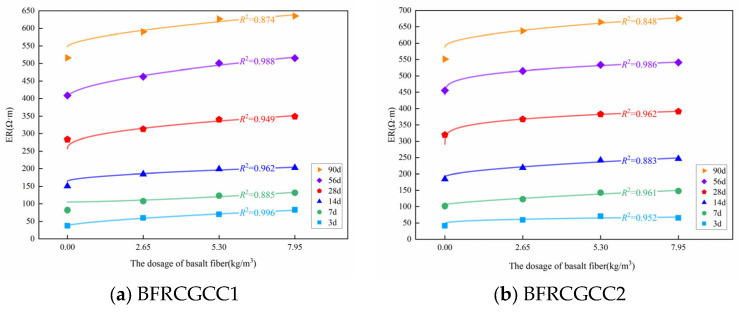
Relationship between ER and fiber dosage at different curing ages.

**Figure 24 materials-18-04627-f024:**
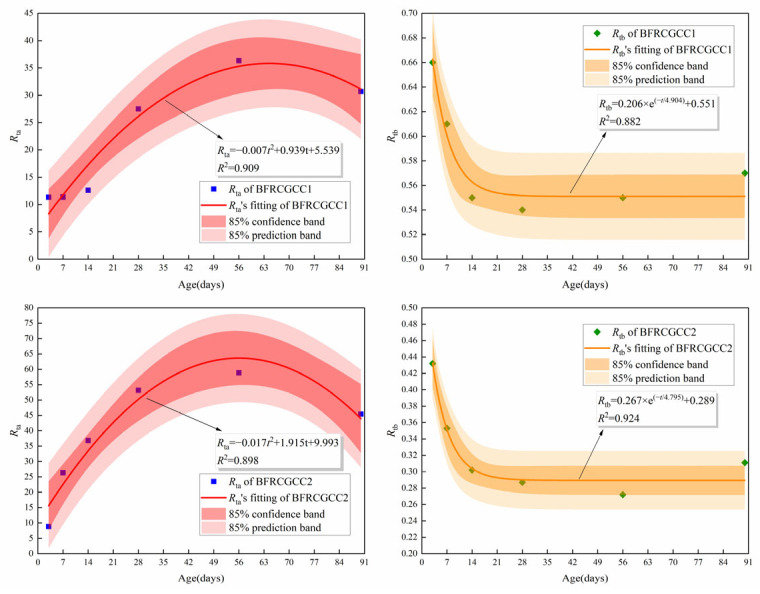
Fitting results of *R*_ta_ and *R*_tb_.

**Figure 25 materials-18-04627-f025:**
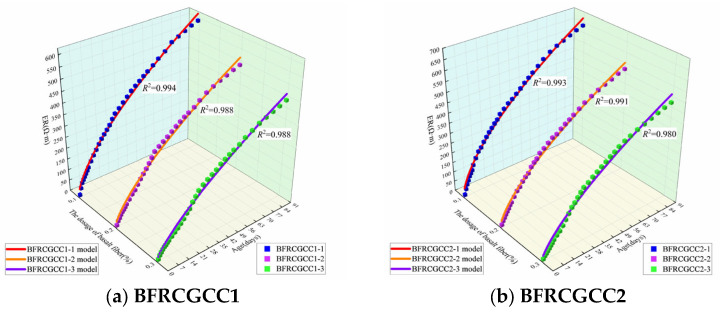
Comparison between experimental values and predicted values of ER.

**Figure 26 materials-18-04627-f026:**
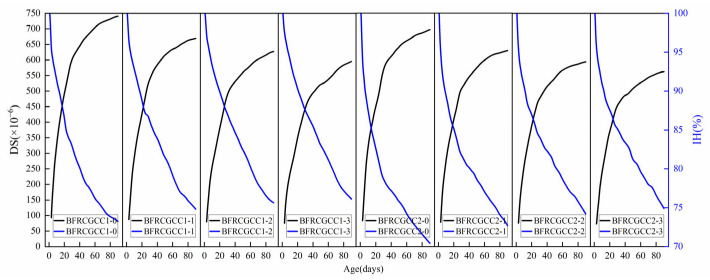
The variations in DS and IH with curing ages.

**Figure 27 materials-18-04627-f027:**
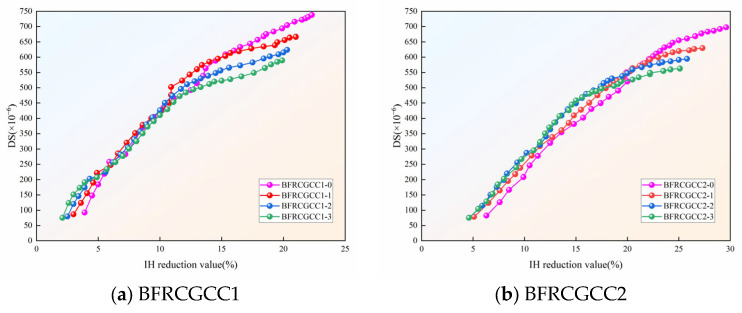
Relationship between the DS and IH reduction in the samples.

**Figure 28 materials-18-04627-f028:**
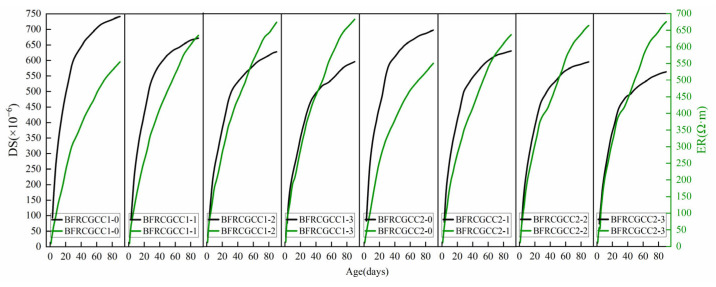
The variation of ER and DS with curing ages.

**Figure 29 materials-18-04627-f029:**
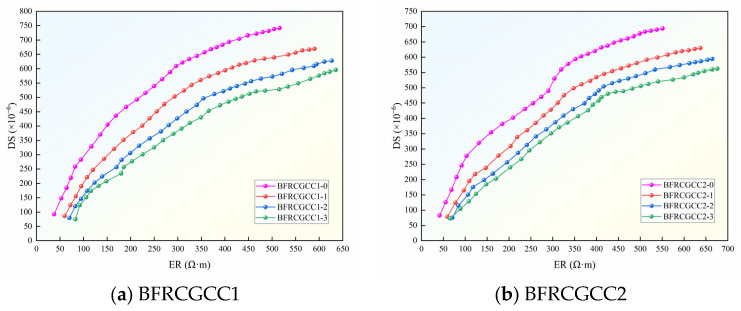
Relationship between ER and DS.

**Figure 30 materials-18-04627-f030:**
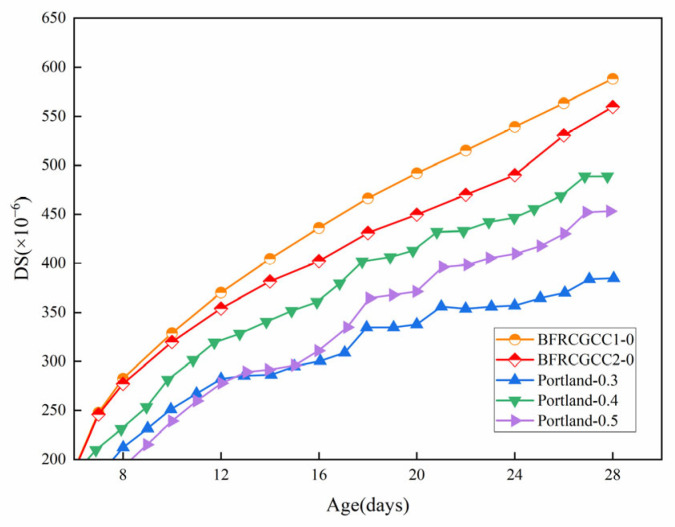
Comparison of measured values with those of Portland cement concrete.

**Table 1 materials-18-04627-t001:** The basic mix proportion of CGCC (kg/m^3^).

Types	Cement	Water	Sand	Coal Gangue Ceramsite	Water Reducer
CGCC1	380	152	653	641	3.8
CGCC2	450	162	623	658	4.5

**Table 2 materials-18-04627-t002:** The properties of cement.

Type	Density/ (kg/m^3^)	Specific Surface Area/ (m^2^/kg)	Particle Size Distribution/μm			Standard Consistency Water Consumption/%	Condensation Time/Min		Compressive Strength/MPa	
			D10	D50	D90		Initial Condensation	Final Condensation	3 Days	28 Days
P·O42.5	3010	343	2.72	16.88	57.42	26.1	181	240	27.9	51.8

**Table 3 materials-18-04627-t003:** The properties of coal gangue ceramsite.

Bulk Density/(kg/m^3^)	Apparent Density/(kg/m^3^)	Cylinder Compressive Strength/MPa	Mud Content/%	Water Absorption Rate/%
844	1321	6.2	<0.1	8.4

**Table 4 materials-18-04627-t004:** Physical properties of the basalt fiber.

Length/mm	Diameter/μm	Density/(kg/m^3^)	Modulus of Elasticity/GPa	Tensile Strength/MPa	Ultimate Elongation/%
35	7~15	2650	100	4200	2~3

**Table 5 materials-18-04627-t005:** Summary of the types of specimens.

Types		Fiber Dosage
BFRCGCC1	BFRCGCC1-0	0
	BFRCGCC1-1	0.1%
	BFRCGCC1-2	0.2%
	BFRCGCC1-3	0.3%
BFRCGCC2	BFRCGCC2-0	0
	BFRCGCC2-1	0.1%
	BFRCGCC2-2	0.2%
	BFRCGCC2-3	0.3%

Note: for the types of BFRCGCCX-Y, X means the water–cement ratio of the matrix, 1—water–cement ratio 0.40, 2—water–cement ratio 0.36; Y means the CGCC with basalt fiber dosage of 0.Y%.

**Table 6 materials-18-04627-t006:** Compressive strength of the specimens with different fiber dosages.

Types	Compressive Strength /MPa	Variation/%	95% Confidence Interval/MPa
BFRCGCC1-0	37.3	--	34.8~37.7
BFRCGCC1-1	36.7	−1.6	
BFRCGCC1-2	35.8	−4.0	
BFRCGCC1-3	35.3	−5.4	
BFRCGCC2-0	51.2	--	49.4~51.3
BFRCGCC2-1	50.4	−1.6	
BFRCGCC2-2	50.1	−2.1	
BFRCGCC2-3	49.8	−2.7	

**Table 7 materials-18-04627-t007:** Regression analysis results of the IH of the CGCC without any reinforcement.

Types	*a*	*b*	*R* ^2^
BFRCGCC1-0	2.38	0.51	0.982
BFRCGCC2-0	4.81	0.41	0.971

**Table 8 materials-18-04627-t008:** Fitting results of *K_t_*.

Types	Parameter	Age/d					
		3	7	14	28	56	90
BFRCGCC1	*K_t_*	0.734	0.926	1.045	1.188	1.369	1.544
	*R* ^2^	0.956	0.926	0.939	0.959	0.992	0.899
BFRCGCC2	*K_t_*	1.21	1.158	1.178	1.405	2.084	2.113
	*R* ^2^	0.889	0.977	0.960	0.864	0.989	0.861

**Table 9 materials-18-04627-t009:** Results of RMSE and MAPE of the samples.

Types	BFRCGCC1-0	BFRCGCC1-1	BFRCGCC1-2	BFRCGCC1-3	BFRCGCC2-0	BFRCGCC2-1	BFRCGCC2-2	BFRCGCC2-3
RMSE	0.834	0.293	0.209	0.165	0.897	0.316	0.348	0.429
MAPE	0.79%	0.28%	0.19%	0.14%	0.93%	0.31%	0.31%	0.39%

**Table 10 materials-18-04627-t010:** Values of the *λ_t_* and *P_t_*.

Types	Coefficients	Age/d					
		3	7	14	28	56	90
BFRCGCC1	*λ_t_*	−530.44	−441.25	−378.93	−327.04	−289.92	−237.43
	*P_t_*	0.934	0.945	0.952	0.963	0.969	0.971
	*R* ^2^	0.916	0.926	0.941	0.927	0.982	0.947
BFRCGCC2	*λ_t_*	−482.24	−418.88	−370.09	−340.46	−268.54	−241.15
	*P_t_*	0.955	0.955	0.970	0.978	0.984	0.991
	*R* ^2^	0.905	0.927	0.950	0.964	0.949	0.961

**Table 11 materials-18-04627-t011:** Results of the RMSE, NRMSE, and MAPE.

Types	BFRCGCC1-0	BFRCGCC1-1	BFRCGCC1-2	BFRCGCC1-3	BFRCGCC2-0	BFRCGCC2-1	BFRCGCC2-2	BFRCGCC2-3
RMSE	15.433 × 10^−6^	11.098 × 10^−6^	12.308 × 10^−6^	15.111 × 10^−6^	19.520 × 10^−6^	14.828 × 10^−6^	11.165 × 10^−6^	9.281 × 10^−6^
NRMSE	0.020	0.016	0.020	0.026	0.027	0.023	0.018	0.016
MAPE	3.78%	2.59%	2.88%	4.50%	4.67%	4.27%	3.29%	2.49%

Note: RMSE is Root Mean Square Error, NRMSE is Normalized Root Mean Square Error, MAPE is Mean Absolute Percentage Error.

**Table 12 materials-18-04627-t012:** Values of *R*_ta_ and *R*_tb_ of different ages.

Types	Coefficients	Age/d					
		3	7	14	28	56	90
BFRCGCC1	*R* _ta_	11.34	11.35	12.61	27.5	36.34	30.69
	*R* _tb_	0.66	0.61	0.55	0.54	0.55	0.57
	*R* ^2^	0.954	0.726	0.816	0.832	0.982	0.811
BFRCGCC2	*R* _ta_	8.82	26.33	36.82	53.16	58.86	45.42
	*R* _tb_	0.432	0.353	0.302	0.287	0.272	0.311
	*R* ^2^	0.772	0.819	0.772	0.945	0.941	0.861

**Table 13 materials-18-04627-t013:** Results of the RMSE, NRMSE, and MAPE of ER.

Types	BFRCGCC1-1	BFRCGCC1-2	BFRCGCC1-3	BFRCGCC2-1	BFRCGCC2-2	BFRCGCC2-3
RMSE	12.779	15.713	13.962	15.784	14.795	20.680
NRMSE	0.022	0.032	0.032	0.025	0.028	0.042
MAPE	1.50%	1.76%	1.34%	1.71%	1.58%	2.19%

**Table 14 materials-18-04627-t014:** Regression results of the relationship between IH reduction and DS.

Types	*A* _1_	*B* _1_	*C* _1_	*R* ^2^
BFRCGCC1-0	−640.40	436.35	1.55	0.997
BFRCGCC1-1	−380.91	349.17	1.70	0.986
BFRCGCC1-2	−462.85	354.41	2.01	0.986
BFRCGCC1-3	−578.60	373.91	3.72	0.985
BFRCGCC2-0	−1533.07	630.83	6.32	0.993
BFRCGCC2-1	−1103.46	511.45	4.50	0.992
BFRCGCC2-2	−624.39	380.87	1.15	0.986
BFRCGCC2-3	−459.27	327.72	0.39	0.982

**Table 15 materials-18-04627-t015:** Fitting results of the relationship between the DS and ER.

Types	*A* _2_	*B* _2_	*C* _2_	*R* ^2^
BFRCGCC1-0	−1210.3	340.13	34.89	0.992
BFRCGCC1-1	−1185.85	304.93	29.34	0.951
BFRCGCC1-2	−1180.7	289.61	27.03	0.998
BFRCGCC1-3	−906.22	244.51	25.09	0.997
BFRCGCC2-0	−1611.41	360.96	71.01	0.993
BFRCGCC2-1	−1452.6	321.53	55.09	0.993
BFRCGCC2-2	−1188.24	280.61	40.53	0.996
BFRCGCC2-3	−890.37	233.90	30.68	0.999

## Data Availability

Data will be made available upon request.

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
