# Peer review of "Investigation on Drying Shrinkage of Basalt Fiber-Reinforced Concrete with Coal Gangue Ceramsite as Coarse Aggregates"

_materials, 2025, doi:10.3390/ma18194627_

Round 1

Reviewer 1 Report

Comments and Suggestions for Authors

The study is carefully conducted and provides a thorough experimental analysis, but its scope is very narrow, focusing on a highly specific material combination (coal gangue ceramsite with basalt fibers). The degree of novelty is therefore modest: while the experimental depth is valuable, the influence of fibers on drying shrinkage and the use of predictive models are already well established, and this work mainly adapts those concepts to a particular case.

Several aspects require improvement:

  • The manuscript is unnecessarily long, with repetitions that reduce clarity; Some figures in addition can be merged to be more compact. Some figures and tables repeat information... etc. 
  • The state of the art in the introduction can be improved and extended for more focused context. 
  • The references show a strong geographical bias with limited representation of international studies; and the level of self-citation is higher than desirable, affecting the balance of the literature review.
  • Furthermore, the English language requires editing, as the text contains redundancies, awkward phrasing, and grammatical inaccuracies.
  • What is the main question addressed by the research?
  • Do you consider the topic original or relevant to the field? Does it address a specific gap in the field? Please also explain why this is/ is not the case. 
  • What does it add to the subject area compared with other published material? 
  • What specific improvements should the authors consider regarding the methodology?  
  • Are the conclusions consistent with the evidence and arguments presented and do they address the main question posed? Please also explain why this is/is not the case. 
  • Are the references appropriate?
  • Any additional comments on the tables and figures. CHECK

Reviewer 2 Report

Comments and Suggestions for Authors

The paper ID materials-3890219 investigates the effects of basalt fiber dosage and matrix strength on the drying shrinkage of coal gangue ceramsite concrete by testing internal humidity, electrical resistivity, and shrinkage behavior over 90 days. A modified AFREM prediction model is proposed, which shows strong agreement with experimental results and offers a practical tool for shrinkage control in engineering applications.

The paper is generally understandable, but many sentences are long, redundant, and awkwardly phrased. Shorter, more concise sentences would improve readability. There are occasional grammar issues (e.g., “specimens regarding to the fiber dosage” should be “specimens with varying fiber dosages”). Consistency in technical terms (e.g., “drying shrinkage (DS)” is sometimes written as “the DS”) should be maintained.

The methods sometimes rely heavily on references to national standards without enough explanation for an international readership.

The rationale behind selecting the specific fiber dosages (0–0.3%) should be justified more clearly.

The results are well-documented with tables and figures, but the text often repeats the data instead of interpreting trends. Some statistical analysis or error ranges could strengthen the findings, especially in comparisons between predicted and experimental results. The explanation of mechanisms (e.g., fiber-matrix interactions) is useful but sometimes speculative without supporting microstructural evidence.

The Conclusions are clear but could be condensed to highlight only the most significant findings and contributions.

The reference list is comprehensive and up-to-date (including works up to 2024–2025). Some citations are Chinese sources (not accessible to all readers); where possible, international equivalents should be included. 

Comments on the Quality of English Language

The paper is generally understandable, but many sentences are long, redundant, and awkwardly phrased. Shorter, more concise sentences would improve readability. There are occasional grammar issues (e.g., “specimens regarding to the fiber dosage” should be “specimens with varying fiber dosages”). Consistency in technical terms (e.g., “drying shrinkage (DS)” is sometimes written as “the DS”) should be maintained.

Reviewer 3 Report

Comments and Suggestions for Authors

Comments in the file

Round 2

Reviewer 1 Report

Comments and Suggestions for Authors

the manuscript can be accepted for publication

Reviewer 2 Report

Comments and Suggestions for Authors

The paper can be accepted.

Reviewer 3 Report

Comments and Suggestions for Authors

The authors have taken all the recommendations into account and made the appropriate changes to the manuscript. I believe it is suitable for publication.